# Chunk, Align, Select: A Simple Long-sequence Processing Method for Transformers

## Abstract

Although dominant in natural language processing, transformer-based models remain challenged by the task of long-sequence processing, because the computational cost of self-attention operations in transformers swells quadratically with the input sequence length. To alleviate the complexity of long-sequence processing, we propose a simple framework to enable the off-the-shelf pre-trained transformers to process much longer sequences, while the computation and memory costs remain growing linearly with the input sequence lengths. More specifically, our method divides each long-sequence input into a batch of chunks, then aligns the inter-chunk information during the encoding steps, and finally selects the most representative hidden states from the encoder for the decoding process. To extract inter-chunk semantic information, we align the start and end token embeddings among chunks in each encoding transformer block. To learn an effective hidden selection policy, we design a dual updating scheme inspired by reinforcement learning, which regards the transformers as environments, and leverages the attention scores and the downstream performance feedback as the rewards to optimize the hidden selection policy. Our empirical results on real-world long-text abstractive summarization and reading comprehension tasks demonstrate effective improvements compared to prior long-sequence processing baselines.

## 1 Introduction

Transformers (Vaswani et al., 2017) have become a fundamental model structure for sequential data modeling (Sun et al., 2019a; Dosovitskiy et al., 2021; Wysocki et al., 2023), especially in Natural Language Processing (NLP) (Devlin et al., 2018; Brown et al., 2020), where texts are regarded as sequences of tokens. Built with transformer architectures, pre-trained language models (PLMs) have recently shown astonishing empirical performance in various NLP tasks such as question answering Yang et al. (2019a), dialogue generation (Byrne et al., 2021), summarization (Rush et al., 2015; Nallapati et al., 2016) and machine translation (Yang et al., 2019b; Pan et al., 2021). However, one fatal weakness, that has hindered transformer-based models from being applied in broader application scenarios, is that the computational consumption of self-attention operations increases quadratically with the input sequence length. Therefore, standard transformers have always been challenged by long text tasks, such as machine reading comprehension (Kwiatkowski et al., 2019; Gong et al., 2020; Pang et al., 2022) and long-text summarization (Huang et al., 2021; Ma et al., 2022).

To enhance transformers' capability in long-sequence processing, prior works pay their attention to two major perspectives, the efficient attention operation and the sub-sequence processing. The efficient attention (Beltagy et al., 2020; Zaheer et al., 2020; Choromanski et al., 2020) aims to reduce the memory and calculation cost of self-attentions while still preserving transformers' empirical performance. Unfortunately, most efficient attention methods require customized self-attention implementations, which are not compatible with existing pre-trained language models. Hence, introducing efficient attention leads to costly further pre-training. Moreover, some empirical results show that sparse-attention methods inevitably sacrifice the performance on short-sequence processing tasks compared with full-attention models (Phang et al., 2022).

Another perspective to long-sequence processing is to decompose the long-input texts into multiple sub-sequences, then process them individually (Moro et al., 2022; Zhang et al., 2022; Mao et al., 2022; Liu et al., 2022). Although utilizing the effectiveness of full-attention blocks, sub-sequence

processing methods are not good at capturing the key semantic information among different chunks. Moreover, for better fusing information across chunks in the decoder, Ivgi et al. (2023) put the same fragment in different chunks, which significantly increases the computational cost.

To gather the advantages of the long-sequence processing methods above, in this paper, we introduce a **Sim**ple learning framework with three typical operations: **C**hunk, **A**lign, and **S**elect, which we call as **SimCAS**. In more detail, SimCAS first chunks the input long text into a sub-sequence batch then feeds the batch through specially designed encoding blocks with an inter-chunk aligning mechanism, and finally selects semantically representative hidden representation via a selection policy. To align the global semantic information among chunks, we introduce a sequential batch alignment (SBA) operation to calibrate the start and end token embeddings with each batch in the encoder layers. For learning the selector, inspired by the recent success of reinforcement learning in NLP (Ouyang et al., 2022), we adopt the Proximal Policy Optimization (PPO) (Schulman et al., 2017) algorithm to train the selector, where the transformer is treated as the environment. Specifically, we employ the calculated attention scores and output feedback to acquire the rewards for action optimization of the selector. To evaluate the effectiveness of our SimCAS, we conducted experiments on four long-document summarization datasets (arXiv, PubMed, GovReport, and SummScreen), two multi-document summarization datasets (Multi-News and WCEP), and one reading comprehension dataset (NarrativeQA). Empirical results show that SimCAS can outperform other long-sequence processing baselines and is highly scalable. Our paper makes three main contributions below.

- We propose a simple framework compatible with existing PLMs for processing long sequences. This significant advancement broadens the range of applications for full-attention PLMs. Notably, unlike some prior works that compromise performance on short-sequence tasks to accommodate long-sequence processing, our method is adept at handling both.
- We posit that the transformer can be conceptualized as a simulation environment for a selection policy. We leverage the transformer's attention and output signals to optimize this policy. This optimized selection policy, in turn, facilitates the transformer to concentrate more effectively on the crucial encoded hidden states.
- Our comprehensive experiments illustrate that SimCAS consistently surpasses previous robust baselines with ease. Furthermore, we provide model scaling, efficiency comparisons, and ablation studies to substantiate the superior performance of our proposed method.

## 2 BACKGROUND

**Language Modeling** The training objective for sequence generation consists of a sequence of token decisions made in an auto-regressive manner. This is formulated as a product of decision probabilities corresponding to specific tokens. Given an input sequence $\boldsymbol{X} = (\boldsymbol{x}_1, \boldsymbol{x}_2, \cdots, \boldsymbol{x}_N)$ and its corresponding output $\boldsymbol{Y} = (\boldsymbol{y}_1, \boldsymbol{y}_2, \cdots, \boldsymbol{y}_M)$, we model the following conditional probability:

$$p_\phi(\boldsymbol{Y}|\boldsymbol{X}) = \prod_{m=1}^M p_\phi(\boldsymbol{y}_m|\boldsymbol{Y}_{<m}, \boldsymbol{X}), \tag{1}$$

where $\boldsymbol{Y}_{<m} = (\boldsymbol{y}_1, \boldsymbol{y}_2, \ldots, \boldsymbol{y}_{m-1})$, and $\phi$ represents the model parameters.

**Proximal Policy Optimization** In the domain of reinforcement learning (RL), Proximal Policy Optimization (PPO) (Schulman et al., 2017) is a widely used policy gradient method (Kakade, 2001) for its remarkable performance and efficiency in solving complex control and decision-making tasks (Vinyals et al., 2019; Akkaya et al., 2019). The vanilla policy gradient estimator has the form: $\nabla_\theta \mathbb{E}_{\pi_\theta(\boldsymbol{a}_t|\boldsymbol{s}_t)}[A_t^\pi(\boldsymbol{a}_t, \boldsymbol{s}_t)] \approx \hat{\mathbb{E}}_t[\nabla_\theta \log \pi_\theta(\boldsymbol{a}_t|\boldsymbol{s}_t)\hat{A}_t]$, where $\boldsymbol{s}_t \in \mathcal{S}$ is the state at $t$-step, $\pi_\theta(\boldsymbol{a}_t|\boldsymbol{s}_t)$ is a stochastic policy acting $\boldsymbol{a}_t$ at $\boldsymbol{s}_t$, $\hat{A}_t$ is the estimated value of the advantage function $A_t^\pi(\boldsymbol{a}_t, \boldsymbol{s}_t)$, and $\hat{\mathbb{E}}_t$ denotes the empirical average over a sample batch. The PPO algorithm improves the training stability of the policy gradient, by optimizing the following objective:

$$L^{CLIP}(\theta) = \hat{\mathbb{E}}_t[\min(r_t(\theta)\hat{A}_t, \text{clip}(r_t(\theta), 1 - \varepsilon, 1 + \varepsilon)\hat{A}_t)], \tag{2}$$

where $r_t(\theta) = \frac{\pi_\theta(\boldsymbol{a}_t|\boldsymbol{s}_t)}{\pi_{\theta_{\text{old}}}(\boldsymbol{a}_t|\boldsymbol{s}_t)}$ is the probability ratio between new and old policies, and $\varepsilon > 0$ is a hyper-parameter for clipping.

## 3 METHODOLOGY

Given a long-input text $\boldsymbol{X} = (\boldsymbol{x}_1, \boldsymbol{x}_2, \ldots, \boldsymbol{x}_N)$ with a fairly large input length $N$, we aim to design a model $p_\phi(\boldsymbol{Y}|\boldsymbol{X})$ to predict a corresponding label sequence $\boldsymbol{Y} = (\boldsymbol{y}_1, \boldsymbol{y}_2, \ldots, \boldsymbol{y}_M)$, where $\boldsymbol{Y}$ can

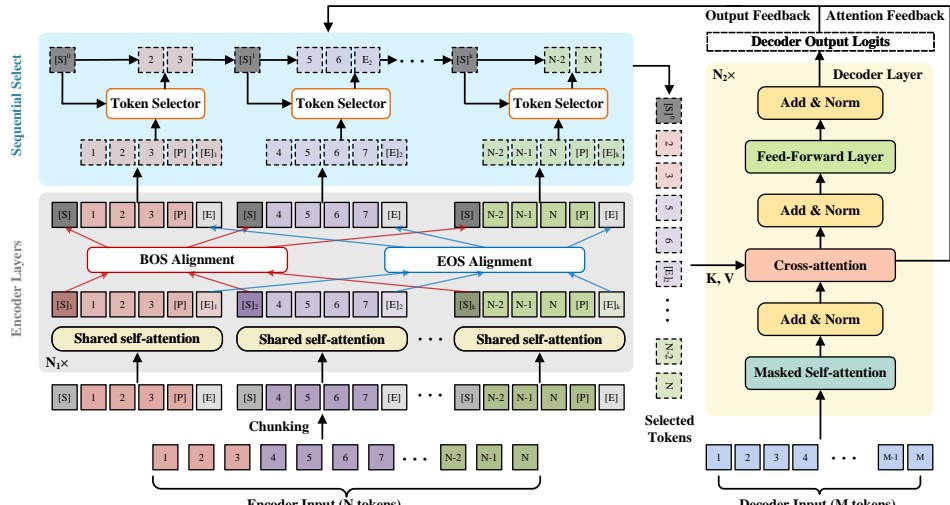

Figure 1: The learning framework of SimCAS: The long inputs are first divided into a batch of chunks, each of which is filled with start token [S], padding token [P] and end token [E]. Then the inter-chunk information can be transferred via the alignment of [S] and [E] representations after each encoder layer. Next, hidden states are selected for decoding steps. The decoder output logits and attention scores are utilized as rewards for updating the token selector.

either be classification labels or output sequence tokens. The major difficulty of the task is that the input length $N$ is so large that the original self-attention operations become infeasible with the quadratic complexity $\mathcal{O}(N^2)$.

To address the challenge, we propose a novel method that intuitively splits the long inputs into chunks with feasible lengths, then selects the most representative tokens for decoding steps. To guarantee inter-chunk semantic information extracted during encoding, we design an aligning scheme in the encoder blocks. In the following, we will describe the chunking scheme, the aligning strategy, and the selector design in detail.

### 3.1 CHUNKING SCHEME

Assume the maximum input sequence length of a pre-trained transformer model is $S$. We first split the long input sequence into $B = \lceil \frac{N}{S} \rceil$ chunks with each chunk length equal to $S$:[1]

$$\left\{ (c_1^k, c_2^k, \ldots, c_S^k) \right\}_{k=1}^{B}, \tag{3}$$

where $c_i^k = x_{(k-1)S+i}$, and $\lceil \cdot \rceil$ is the ceiling function. Since the chunk might not have $S$ tokens from $X$, we append special padding tokens at the end (see Figure 1). After chunking, we add additional *beginning of sentence* ([S]) and *end of sentence* ([E]) tokens to each chunk, and treat the chunks as a normal transformer input batch $\mathcal{C}$ with batch size $B$:

$$\mathcal{C} = \left\{ ([S], c_1^k, c_2^k \ldots, c_S^k, [E]) \right\}_{k=1}^{B}. \tag{4}$$

### 3.2 SEQUENTIAL BATCH ALIGNMENT

After chunking the input text into a standard token batch, we can encode it with the transformer encoder layers. As in Figure 1, we assume the encoder has $N_1$ layers. Denote the hidden representations of $k$-th chunk in $\mathcal{C}$ (in equation 4) at $l$-th encoder layer as $\boldsymbol{H}^{k,l} = (\boldsymbol{h}_0^{k,l}, \boldsymbol{h}_1^{k,l}, \ldots, \boldsymbol{h}_S^{k,l}, \boldsymbol{h}_{S+1}^{k,l})$, where $\boldsymbol{h}_0^{k,l}$ and $\boldsymbol{h}_{S+1}^{k,l}$ are hidden representation of [S] and [E] tokens respectively, and $\boldsymbol{h}_i^{k,l}$ is the embedding for $c_i^k$ with $1 \leq i \leq S$.

---

[1]To simplify the expression, we assume that a sentence is in only one chunk. In the practical experiment, we use the sentence-level segmentation (Moro & Ragazzi, 2022) to acquires a sequence of sentences, and then employ the greedy method to allocate sequentially these sentences to the chunks. If there is not enough space in the chunk for the current sentence, we will use the padding tokens to fill the chunk.

As mentioned in Section 1, chunk-level methods make it difficult to capture the inter-chunk global semantic information. Inspired by recent work using special tokens for full sequence information (Mohtashami & Jaggi, 2023), we design a sequential batch alignment, which aligns the information of [S] and [E] tokens at each encoding block, for [S] and [E] have been well-recognized representative of sentence semantics (Devlin et al., 2018). More specifically, at $l$-th layer, our batch alignment average the hidden states of [S] and [E] of all chunks:

$$\bar{h}_{\mathrm{BOS}}^l = \frac{1}{B} \sum_{k=1}^B h_0^{k,l}, \quad \bar{h}_{\mathrm{EOS}}^l = \frac{1}{B} \sum_{k=1}^B h_{S+1}^{k,l}. \tag{5}$$

Then we replace $h_0^{k,l}$ and $h_{S+1}^{k,l}$ with the aligned $\bar{h}_{\mathrm{BOS}}^l$ and $\bar{h}_{\mathrm{EOS}}^l$ into the hidden states for the next-layer encoding block, as shown in Figure 1.

### 3.3 TOKEN SELECTOR

After being encoded into the last hidden space with our sequential batch aligning scheme, the chunks should be reformed back to a sequence for the next decoding steps. Directly tiling the chunks' representations back into a sequence is still infeasible because the overlong sequence makes it difficult to fuse information at the decoding stage. Therefore, we propose a token selector to select the most representative hidden representations for decoding steps. Inspired by Ramamurthy et al. (2023), we design the selector from the perspective of reinforcement learning (RL).

**Selection Module Design** Formally, the token selector takes the last hidden states $H^L = \{h_0^{k,L}, h_1^{k,L}, \dots, h_S^{k,L}, h_{S+1}^{k,L}\}_{k=1}^B$ and selection actions $A_t = (a_1, a_2, \dots, a_{t-1})$ as inputs and predicts the next selection action $a_t$, where each action $a_t$ has two values *"select"* and *"skip"* for operating the $t$-th token $x_t$ in $X$. We set the state of RL as $s_t = (H^L, A_t)$, then the selector is a policy $\pi_\theta(a_t|s_t)$ to predict next action $a_t$.

We implement the selector with the actor-critic style (Konda & Tsitsiklis, 1999). Both actor and critic are simple feed-forward networks, but the actor outputs a probability distribution over the action space and the critic outputs a single scalar value. At state $s_t$, to consider the sequential effect of previous action $A_t$, we first take the average of all selected hidden states as

$$\bar{h}_t = \frac{\sum_{k,i} \mathbb{I}_{\{a_j = "select", j < t\}} h_i^{k,L}}{\sum_{k,i} \mathbb{I}_{\{a_j = "select", j < t\}}}, \tag{6}$$

where $j = (k-1)S + i$ maps chunk indices back to the input sequence, and $\mathbb{I}_{\{\cdot\}}$ is the indicator function. Then we concatenate the current selector state $\bar{h}_t$ and token information $h_i^{k,L}$, $t = (k-1)S + i$, to predict next action $a_t$ via the actor:

$$\pi_\theta(a_t|s_t) = \mathrm{actor}(\bar{h}_t, h_i^{k,L}). \tag{7}$$

**Reward Design** To train the selector within an RL scheme, we treat the transformer as an environment and design action rewards. Inspired by Yuan et al. (2021), we can directly utilize the language modeling likelihood as the generation quality reward for selection actions:

$$R_{\mathrm{LM}} = \xi \exp\{\tfrac{1}{M} \log p_\phi(Y|X)\}, \tag{8}$$

where $\xi$ is a coefficient that magnifies the value of the reward for easier optimization of the selector. However, $R_{\mathrm{LM}}$ is only a scalar value, which cannot provide fine-grained guidance to the selector. Therefore, we use the input-output cross-attention scores to calibrate $R_{\mathrm{LM}}$. More specifically, we denote the cross-attention matrix of the $q$-th attention head in the $l$-th layer of the decoder is denoted as $A_q^l \in \mathbb{R}^{M \times N}$, and the overall cross-attention

$$\bar{A} = \tfrac{1}{N_2 \cdot Q} \sum_{l=1}^{N_2} \sum_{q=1}^Q A_q^l, \tag{9}$$

where $N_2$ is the decoder layer number, and $Q$ is the cross-attention head number. With the overall feedback $R_{\mathrm{LM}}$ and cross-attention $\bar{A} = (\bar{a}_{ij})_{M \times N}$, we adjust the reward $R_j^+$ to each selected tokens. For *"skip"* action, we intend to limit the selected sequence length. Assume the number of overall input tokens and selected token representations is $L_{\mathrm{all}}$ and $L_{\mathrm{select}}$ respectively. We set a hyper-parameter $L_{\mathrm{hyper}}$ to control the size of $L_{\mathrm{select}}$ with the skipping reward $R^-$:

$$R_j^+ = \frac{\bar{a}_j}{1 - \bar{a}_0} R_{\mathrm{LM}}, \quad \bar{a}_j = \frac{1}{M} \sum_{i=1}^M \bar{a}_{ij}, \quad R^- = \begin{cases} \frac{R_{\mathrm{LM}}}{L_{\mathrm{all}}} & \text{if } L_{\mathrm{select}} < L_{\mathrm{hyper}} \\ \frac{R_{\mathrm{LM}}}{L_{\mathrm{select}}} & \text{otherwise.} \end{cases} \tag{10}$$

With the selector and rewards designed above, we can optimize the selector with the PPO algorithm described in Section 2. Note that in our setups, the environment (the transformer) is changing during the training steps. Therefore, we alternatively update the selector and transformer: in each interaction, we first fix the transformer and use the reward $R_{\mathrm{LM}}$ and cross-attention scores to update the selector, then fix the selector and update the transformer with language modeling loss. In addition, the selector is similar to the chunk-wise RNN, so the time overhead of the selection process is low.

## 4 RELATED WORK

**Efficient Transformers** The attention mechanism in transformers requires quadratically increased computational complexity with respect to the input sequence length, which limited the application scenarios, especially for long-text processing. To address this issue, various previous works have been proposed for designing more efficient attention operations (Tay et al., 2023; Fournier et al., 2023). Longformer (Beltagy et al., 2020), BIGBIRD (Zaheer et al., 2020), GMAT (Gupta & Berant, 2020), and ETC (Ainslie et al., 2020) reduce the memory consumption of dense attentions by elaborate combinations of global attention and local attention mechanisms. Reformer (Kitaev et al., 2020) leverages a locality-sensitive hashing to the attention mechanism, changing its complexity from $\mathcal{O}(n^2)$ to $\mathcal{O}(n \log n)$, where $n$ is the input text sequence length. Routing Transformer (Roy et al., 2021) applies a sparse routing module based on online k-means to self-attention while reducing the overall complexity of attention. Approximation-based methods, such as Performers (Choromanski et al., 2020) and RFA (Peng et al., 2021), use linear space and time complexity to estimate the attention matrix based on random features. Luna (Ma et al., 2021) attends only to a fixed number of hidden vectors. Linformer (Wang et al., 2020) calculates self-attention by a low-rank matrix. However, the vast majority of these methods are difficult to apply to existing PLMs. Moreover, Xiong et al. (2022) proposes that many efficient-attention transformers do not even perform as well as simple local-attention models on downstream language tasks.

**Chunking Methods for Long Sequence** Another solution for long-sequence processing is to perform sequence chunking and then process them respectively (Zhong et al., 2022; Wang et al., 2023). Among chunking methods, SLED (Ivgi et al., 2023) splits the long sequence into overlapping chunks and processes each chunk with the encoder, then fuses cross-chunk information with the decoder. PageSum (Liu et al., 2022) separates the long sequence into non-overlapping chunks and effectively tackles them by the principle of locality (Denning, 2005). Unlimiformer (Bertsch et al., 2023) encodes long inputs in chunks and utilizes only the top-k input tokens for every attention head.

**Length Extrapolation** Length extrapolation in transformers refers to their ability to handle input sequences of varying lengths during both training and inference (Press et al., 2022; Sun et al., 2023). Transformers use a self-attention mechanism to capture dependencies across positions in a sequence, allowing them to generalize well to sequences of different lengths. This flexibility is essential for tasks like natural language processing and time series analysis, where input lengths can vary.

**Sequence Length Reduction** Reducing the length of hidden states (Guan et al., 2022; Kim et al., 2022) is the method of model compression from the width perspective, which is promising since some studies showed that there is redundant encoded information in token representations (Ethayarajh, 2019; Klafka & Ettinger, 2020). Among the redundancy, some tokens carry more task-specific information than others, suggesting that these tokens are more salient and imperative to be selected to feed into subsequent operations. Compared with model compression via layer-wise pruning, token-level pruning does not come at the expense of model performance in complex reasoning (Sanh et al., 2019; Sun et al., 2019b).

## 5 EXPERIMENTS

To evaluate the performance of SimCAS, we conduct experiments on the long-text abstractive summarization and machine reading comprehension tasks. In the following, we introduce the information about the datasets, baselines, model implementations, and evaluation results of our experiments.

### 5.1 DATASETS

We conduct experiments on two types of NLP tasks: long-text summarization and machine reading comprehension. For long-text summarization, we use four single-document summarization datasets: arXiv, PubMed (Cohan et al., 2018), GovReport (Huang et al., 2021), SummScreen (Chen

Table 1: Average results on arXiv and PubMed test sets. R-1/2/L is the ROUGE-1/2/L F1 score. BS refers to the neural model-based metrics BERTScore. The best results are bolded.

| Base Model | Method | arXiv | | | | PubMed | | | |
|---|---|---|---|---|---|---|---|---|---|
| | | R-1 | R-2 | R-L | BS | R-1 | R-2 | R-L | BS |
| LED$_{large}$ | Standard | 46.63 | 19.62 | 41.83 | - | - | - | - | - |
| LED$_{large}$ | PRIMERA | 47.60 | **20.80** | 42.60 | - | - | - | - | - |
| PEGASUS$_{large}$ | Standard | 44.21 | 16.95 | 38.83 | - | 45.97 | 20.15 | 41.34 | - |
| PEGASUS$_{large}$ | BIGBIRD | 46.63 | 19.02 | 41.77 | - | 46.32 | 20.65 | 42.33 | - |
| BART$_{large}$ | HEPOS | 47.87 | 20.00 | 41.50 | - | 47.93 | 20.74 | 42.58 | - |
| BART$_{base}$ | Standard | 40.36 | 13.78 | 36.11 | 59.44 | 40.36 | 13.29 | 35.02 | 61.77 |
| BART$_{large}$ | Standard | 42.97 | 15.54 | 37.02 | 61.18 | 42.87 | 15.44 | 36.93 | 63.08 |
| BART$_{base}$ | SimCAS | 47.22 | 19.35 | 42.25 | 63.51 | 48.17 | 21.11 | 43.90 | 66.33 |
| BART$_{large}$ | SimCAS | **48.14** | 19.77 | **42.93** | **63.78** | **48.65**$^†$ | **21.40** | **44.14** | **66.52** |

et al., 2022), and two multi-document summarization datasets: Multi-News (Fabbri et al., 2019b), WCEP (Gholipour Ghalandari et al., 2020). For the reading comprehension task, we test on the NarrativeQA (Kočiský et al., 2018) dataset.

**arXiv & PubMed**[2] are two long-document summarization datasets in the scientific research domain. Each document is a scientific paper whose summary is the corresponding abstract.

**GovReport**[3] is a long-document summarization dataset based on reports published by the U.S. Government Accountability Office and Congressional Research Service.

**SummScreen**[4], which includes TV series transcripts, often presents plot details in an indirect manner through character dialogues scattered throughout the transcript. These details need to be identified and consolidated to create concise plot descriptions.

**Multi-News**[5] is a large-scale multi-document summarization dataset. It consists of news articles and human-written summaries of these articles. Each summary is professionally written by editors and with links to the original articles cited.

**WCEP**[6] is a dataset for multi-document summarization (MDS). It contains short, human-written summaries about news events, obtained from the Wikipedia Current Events Portal (WCEP).

**NarrativeQA**[7] is a reading comprehension dataset over entire books from Project Gutenberg and movie scripts from different websites.

More detailed information about the datasets is provided in Appendix §A.

## 5.2 BASELINES

There are several baselines for comparison: HiMAP (Fabbri et al., 2019a), BERTREG (Gholipour Ghalandari et al., 2020), Submodular+Abs (Gholipour Ghalandari et al., 2020), BART (Lewis et al., 2020), PEGASUS (Zhang et al., 2020), DynE (Hokamp et al., 2020), Graph-Sum (Li et al., 2020), BART-Long-Graph (Pasunuru et al., 2021), HEPOS (Huang et al., 2021), SLED (Ivgi et al., 2023), Memorizing Transformers (Wu et al., 2022), Unlimiformer (Bertsch et al., 2023), LED (Longformer Encoder-Decoder) (Beltagy et al., 2020), BIGBIRD (Zaheer et al., 2020), PRIMERA (Xiao et al., 2022), and LED+RELAX (Parnell et al., 2022). More details of baselines can be found in Appendix §B.

## 5.3 IMPLEMENTATION DETAILS

Our implementation is based on *PyTorch* (Paszke et al., 2019) and *Transformers* (Wolf et al., 2020) libraries. We train our model by using 8 NVIDIA V100 32G GPUs.

---

[2]https://github.com/armancohan/long-summarization

[3]https://github.com/luyang-huang96/LongDocSum

[4]https://github.com/mingdachen/SummScreen

[5]https://github.com/Alex-Fabbri/Multi-News

[6]https://github.com/allenai/PRIMER

[7]https://github.com/deepmind/narrativeqa

Table 2: Average results on GovReport and SummScreen test sets. R-1/2/L is the ROUGE-1/2/L F1 score. BS refers to the neural model-based metrics BERTScore. The best results are bolded.

| Base Model | Method | GovReport | | | | SummScreen | | | |
|---|---|---|---|---|---|---|---|---|---|
| | | R-1 | R-2 | R-L | BS | R-1 | R-2 | R-L | BS |
| BART$_{base}$ | SLED | 54.70 | 24.40 | 25.40 | - | 32.70 | 7.90 | 19.10 | - |
| BART$_{large}$ | SLED | 57.50 | 26.30 | 27.40 | - | 35.20 | 8.70 | 19.40 | - |
| LED$_{large}$ | PRIMERA | 55.10 | 23.90 | 25.90 | 67.00 | 32.30 | 7.10 | 18.30 | 57.10 |
| BART$_{base}$ | Memorizing | 55.20 | 25.10 | 26.40 | 67.50 | 32.70 | 7.40 | 19.20 | 57.40 |
| BART$_{base}$ | Unlimiformer | 56.60 | 26.30 | 27.60 | 68.20 | 34.70 | 8.50 | **19.90** | 58.50 |
| PRIMERA | Unlimiformer | 57.40 | 26.20 | **28.00** | 68.10 | 33.30 | 7.60 | 18.90 | 57.70 |
| BART$_{base}$ | Standard | 51.72 | 19.37 | 23.11 | 64.12 | 29.73 | 5.23 | 15.65 | 54.30 |
| BART$_{large}$ | Standard | 52.94 | 19.78 | 23.71 | 64.44 | 29.89 | 5.32 | 15.71 | 54.43 |
| BART$_{base}$ | SimCAS | 59.30 | 25.95 | 27.07 | 68.17 | 43.45 | 12.74 | 18.38 | 62.46 |
| BART$_{large}$ | SimCAS | **60.29** | **26.68** | 27.97 | **68.64** | **44.15** | **13.42** | 18.50 | **62.82** |

For a fair comparison, we uniformly use the full-attention PLM BART[8] on the seven public datasets above. Built on the BART, our framework introduces an additional parameterized selector to focus on more task-specific token representations. The selector follows the actor-critic style (Konda & Tsitsiklis, 1999) and contains around 8M parameters. There are two Adam optimizers with $\beta_1 = 0.9$, $\beta_2 = 0.999$ for BART and selector respectively.

Additionally, to handle longer sequences during the training phase, we set the chunk size to 512 instead of 1024 in all experiments (considering the start token and end token). To maintain chunk-wise alignment, we pad the chunks to uniform the size of each chunk. During the forward propagation, the embedding layer embeds position representations for each chunk independently.[9]

During the training phase, due to memory limitation, maximum input lengths for BART$_{large}$ and BART$_{base}$ are set to 8192 and 16384 respectively, unless otherwise specified. For efficient training, we update the parameters of the original backbone and selector alternatively. The reward estimation of each action is computed based on decoder cross-attention and the output feedback of the generative model. This estimation process is detached from the computation graph and does not participate in backpropagation.

At the inference stage, compared to the original generation process, our framework only adds a chunk-wise selection procedure between the encoder and the decoder, which takes very little time. At the decoding stage, the target sequence is generated with beam search in an auto-regressive manner (Wiseman & Rush, 2016).

## 5.4 EVALUATIONS

Like most previous works, for abstractive summarization tasks, we measure the quality of generated summaries using the popular metric ROUGE (Lin, 2004). On the test set of arXiv, PubMed, GovReport, SummScreen, Multi-News, and WCEP, we report full-length F1-based ROUGE-1, ROUGE-2, and ROUGE-L scores computed with the standard ROUGE Perl package. Furthermore, we also use a popular model-based semantic metric BERTScore[10] (Zhang* et al., 2020) to demonstrate the superiority of our approaches comprehensively.

As for the reading comprehension task, we use the F1 and exact match (EM) metrics defined in SCROLLS (Shaham et al., 2022) to evaluate the model performance on the NarrativeQA dataset.

---

[8]The checkpoints of BART are "facebook/bart-base" and "facebook/bart-large" containing around 139 M and 406 M parameters respectively, whose maximum encoding length is 1024.

[9]Considering the context of each chunk is different, although our design may lead to repeated position representation, we argue that after the encoding stage, the combination of the same token and position information would still produce various representations (Kazemnejad et al., 2023)

[10]In order to make a fair comparison with the previous work, we also use checkpoint "facebook/bart-large-mnli" for BERTScore.

Table 3: Average results on Multi-News (left) and WCEP (right) test sets. R-1/2/L is the ROUGE-1/2/L F1 score. BS refers to the model-based metrics BERTScore. The best results are bolded.

| System | R-1 | R-2 | R-L | BS |
|---|---|---|---|---|
| BART$_{large}$ | 42.04 | 14.88 | 23.34 | - |
| HiMAP | 44.17 | 16.05 | 21.38 | - |
| GraphSum | 45.87 | 17.56 | 23.39 | - |
| BART-Long-Graph | 49.24 | 18.99 | 23.97 | - |
| LED+RELAX | 47.23 | 18.86 | 25.03 | - |
| PRIMERA | **49.94** | **21.05** | 25.85 | - |
| BART$_{large}$ | 42.16 | 14.69 | 23.51 | 60.70 |
| BART$_{base}$+SimCAS | 48.88 | 20.01 | 25.32 | 65.05 |
| BART$_{large}$+SimCAS | 49.40$^{\dagger}$ | 20.47 | **25.96** | **65.40** |

| System | R-1 | R-2 | R-L | BS |
|---|---|---|---|---|
| BERTREG | 35.00 | 13.50 | 25.50 | - |
| SUBMODULAR+ABS | 34.40 | 13.10 | 25.00 | - |
| DynE | 35.40 | 15.10 | 25.60 | - |
| LED | 39.79 | 18.94 | 32.10 | - |
| LED+RELAX | 41.11 | 19.46 | 33.13 | - |
| PRIMERA | 46.08 | **25.21** | 37.86 | - |
| BART$_{large}$ | 37.66 | 15.98 | 31.01 | 62.32 |
| BART$_{base}$+SimCAS | 45.68 | 22.80 | 37.71 | 70.59 |
| BART$_{large}$+SimCAS | **46.29** | 24.45 | **38.61** | **71.38** |

Both metrics normalize the reference and system output strings via lowercasing, removing punctuation and stopwords, and normalizing whitespace.

# 6 DISCUSSION

## 6.1 RESULTS

Table 1 and 2 report results over four long-document test sets. We note that casting the backbone BART$_{base}$ into our SimCAS can significantly improve the model performance, and increasing the size of the model can further improve performance. Our approach outperforms baseline on several metrics and achieves new state-of-the-art performance on the PubMed test set. In Table 3, the results of multi-document summarization tasks have a similar trend. Apart from PRIMERA, which customizes a pre-training objective for multi-document summarization, our method significantly outperforms previous results. See Figure 2, we observe a substantial improvement in the performance of our model on the NarrativeQA test set compared to previous works.

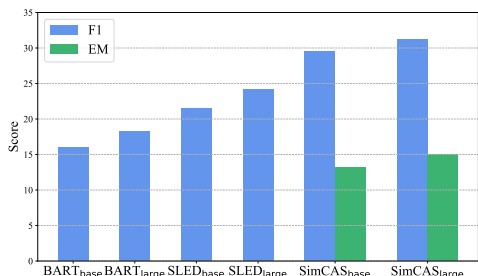

Figure 2: System performance comparison on NarrativeQA test set.

Given the fact that this dataset has an extremely long average input length ($\geq$100K) and short average output length ($\leq$10), we attribute this significant performance enhancement to our method's proficiency in filtering out an immense amount of task-irrelevant information. This, in turn, enables the decoding blocks to fuse information more efficiently.

## 6.2 ANALYSIS

We further analyze the properties of our framework to gain more insights.

**Ablation Study** SimCAS is composed of three components: `Chunk`, `Align`, and `Select`. To investigate the contributions of each component, we independently removed each one. Notably, after removing `Chunk`, the maximum input length was restricted to 1024, causing `Align` to fail. As demonstrated in Table 4, the performance experienced a significant decline when either `Chunk` or `Select` was removed, underscoring their effectiveness. The `Align` component improved performance on all datasets, with the exception of the Multi-News dataset. We hypothesize that more sophisticated alignment methods could potentially enhance performance further.

**Effectiveness of Conditional Skipping Reward** In practice, to circumvent the selector selecting exceedingly many token representations, we design a conditional reward mechanism for skipping actions to soft constrain the number of selected representations. Figure 3 (left) exhibits the effectiveness of our reward mechanism. We can see that as the training progresses, the number of selected tokens gradually converges to the threshold (2048).

**Increasing Maximum Input Length** We explore the performance change of our model on a sin-

Table 4: Ablation study results on the development sets of all datasets. Performance changes compared with the full model (BART$_{base}$+SimCAS) are reported. The metrics of summarization and reading comprehension datasets are the average of ROUGE-1/2/L and F1 respectively.

| Datasets | w/o Chunk | w/o Align | w/o Select | SimCAS | $\Delta_{Chunk}$ | $\Delta_{Align}$ | $\Delta_{Select}$ |
|---|---|---|---|---|---|---|---|
| arXiv | $30.10 \pm 0.73$ | $36.12 \pm 0.47$ | $32.70 \pm 0.62$ | $\mathbf{36.46 \pm 0.56}$ | ↑ 21.13% | ↑ 0.94% | ↑ 11.50% |
| PubMed | $29.63 \pm 0.51$ | $37.20 \pm 0.64$ | $34.32 \pm 0.59$ | $\mathbf{37.77 \pm 0.51}$ | ↑ 27.47% | ↑ 1.53% | ↑ 10.05% |
| GovReport | $31.23 \pm 0.50$ | $37.12 \pm 0.66$ | $32.91 \pm 0.81$ | $\mathbf{37.54 \pm 0.48}$ | ↑ 20.20% | ↑ 1.13% | ↑ 14.07% |
| SummScreen | $19.71 \pm 0.49$ | $20.69 \pm 0.45$ | $19.63 \pm 0.62$ | $\mathbf{25.15 \pm 0.42}$ | ↑ 27.60% | ↑ 2.56% | ↑ 08.10% |
| Multi-News | $25.34 \pm 0.38$ | $31.45 \pm 0.55$ | $27.98 \pm 0.35$ | $\mathbf{31.36 \pm 0.51}$ | ↑ 23.76% | ↓ 0.29% | ↑ 12.08% |
| WCEP | $28.42 \pm 0.72$ | $35.01 \pm 0.64$ | $30.33 \pm 0.76$ | $\mathbf{35.31 \pm 0.59}$ | ↑ 24.24% | ↑ 0.86% | ↑ 16.42% |
| NarrativeQA | $21.70 \pm 0.55$ | $31.52 \pm 0.46$ | $23.20 \pm 0.23$ | $\mathbf{31.76 \pm 0.44}$ | ↑ 46.36% | ↑ 0.76% | ↑ 36.90% |

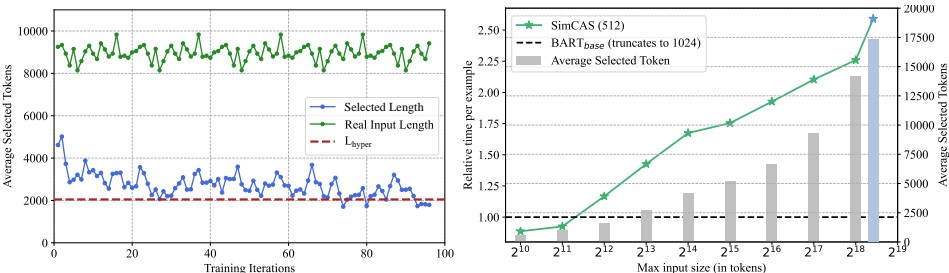

Figure 3: The left-hand-side plot shows the change of the actual number of tokens entered into the model and the number of tokens selected by our selector during training. The red dashed line represents the conditional boundary for the skipping reward. The right-hand-side plot shows the effect of increasing the number of input tokens in the inference phase on the time latency and the number of selected tokens. The area marked in blue on the right represents the limit of the number of tokens that the V100 can handle (350K tokens).

gle 32G V100 GPU with the increase of input sequence length during inference. First, we test the efficiency of our framework in processing long sequences. In our setup, we use BART$_{base}$ as the backbone. Figure 3 (right) shows that in the case of small magnitude, with the exponential increase of input length, the time cost still increases approximately linearly. We believe that this phenomenon occurs because our method significantly reduces the computation complexity and short sequences make insufficient use of the computation cores. We also note that processing the sequence containing 350K tokens in the inference phase approaches the limit of memory. Additionally, we notice that even as the number of input tokens increases dramatically, the selected hidden states are kept at a reasonable size, indicating the capability of our selector to ignore low-contribution information.

## 7 CONCLUSION AND FUTURE WORK

In this paper, we introduced a simple method for long-text processing via chunking, aligning, and selecting, called SimCAS. We divided the input long sequence into chunks, and encoded them with sequential batch alignment to capture the inter-chunk semantics. To select the important token representations in the encoded output of the encoder, we introduced a reinforcement learning-based token selector with the PPO method. We leverage the transformer as an environment and design a reward scheme for the corresponding actions based on the output logits and decoder cross-attention feedback to optimize the hidden token selector. Substantial experiment results and analyses demonstrate the satisfying effectiveness of SimCAS. Besides, our method does not depend on any particular tasks or models, which have good generalization ability for various application scenarios.

For future work, our method can be naturally adopted into non-language long-sequence processing tasks, such as molecular structure analysis. Also, SimCAS has the potential to enhance the long text pre-training to transformers.

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

# A    DATASET STATISTICS

Table 5: Statistics of used datasets.

| Dataset | Train | Valid | Test | Avg. Input Tokens |
|---|---|---|---|---|
| arXiv | 203.0K | 6.4K | 6.4K | 6.0K |
| PubMed | 119.9K | 6.6K | 6.7K | 3.0K |
| GovReport | 17.5K | 1.0K | 1.0K | 9.6K |
| SummScreen | 22.6K | 2.1K | 2.1K | 6.6K |
| Multi-News | 44.9K | 5.6K | 5.6K | 1.8K |
| WCEP | 8.1K | 1.0K | 1.0K | 3.9K |
| NarrativeQA | 32.7K | 3.5K | 10.6K | 121.7K |

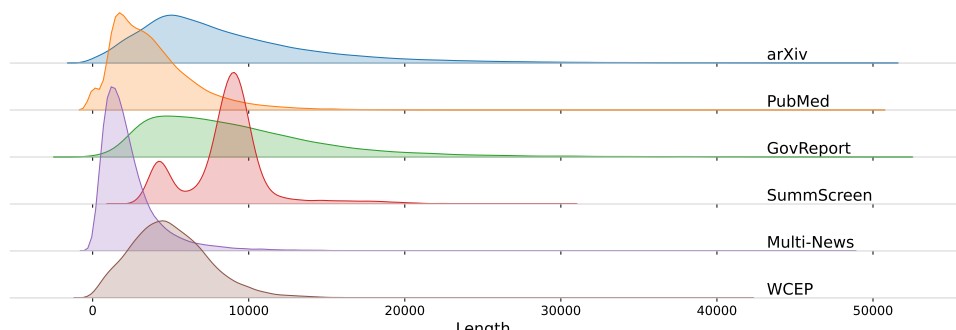

Figure 4: Input text length distributions on the six summarization datasets

# B    INTRODUCTION FOR BASELINES

We use the competitive baselines that demonstrate the downstream task results for comparison. Among them, **BART** (Lewis et al., 2020) is a standard full-attention PLM for sequence generation. Compared with BART, **PEGASUS** (Zhang et al., 2020) has a tailored pre-training objective for abstractive text summarization. **LED** (Longformer Encoder-Decoder) (Beltagy et al., 2020) uses the sparse attention-based encoder and full-attention decoder. Before pre-training, its parameters are initialized from BART. **BIGBIRD** (Zaheer et al., 2020), for an encoder-decoder setup, also introduces their specified sparse attention mechanism only at the encoder side. **PRIMERA** based on LED introduces a task-specific per-training objective for multi-document summarization. **SLED** (Ivgi et al., 2023) processes long sequences via short-context PLMs. The origin long sequence is partitioned into overlapping chunks. **HEPOS** (Huang et al., 2021) proposes head-wise positional strides to effectively pinpoint salient information from the source. **Memorizing Transformers** (Wu et al., 2022) employs a trainable attention gate to moderate between the standard cross-attention and attention over retrieved keys from a datastore. **Unlimiformer** (Bertsch et al., 2023) uses the k-nearest-neighbor KNN index to choose full-length input to reduce computation overhead. **HiMAP** (Fabbri et al., 2019a) expands the existing pointer-generator network model into a hierarchical network. **GraphSum** Li et al. (2020) employs well-known graph representations of documents to effectively process multiple input documents for abstractive summarization. **BART-Long-Graph** (Pasunuru et al., 2021) is fine-tuned based on LED and additionally injects discourse graph. **LED+RELAX** Parnell et al. (2022) introduces a RELAX gradient estimator with multi-document coverage reward. **BERTREG** (Gholipour Ghalandari et al., 2020) is a regression-based sentence ranking system with BERT embedding, which is used as an extractive summarization method. **Submodular+Abs** (Gholipour Ghalandari et al., 2020) consists of a submodular-based extractive summarizer and a bottom-up abstractive summarizer. **DynE** (Hokamp et al., 2020) ensembles multiple-document for abstractive summarization by single document summarization models.

Table 6: Perplexity on all seven test sets including arXiv, PubMed, GovReport, SummScreen, Multi-News, WCEP, and NarrativeQA.

| Method | arXiv | PubMed | GovReport | SummScreen | Multi-News | WCEP | NarrativeQA |
|--------|-------|--------|-----------|------------|------------|------|-------------|
| BART | 31.82 | 34.81 | 42.10 | 48.91 | 49.40 | 50.91 | 23.34 |
| SimCAS | 6.89 | 5.58 | 6.23 | 10.80 | 8.25 | 8.85 | 4.71 |

Table 7: The average time overhead of the selection process (Select) and the whole process (All), and their ratio (Ratio) for a single sample.

| Method | arXiv | PubMed | GovReport | SummScreen | Multi-News | WCEP | NarrativeQA |
|--------|-------|--------|-----------|------------|------------|------|-------------|
| Selection (ms) | 157.5 | 70.4 | 293.9 | 103.4 | 74.2 | 9.8 | 112.4 |
| All (ms) | 9661.5 | 13799.1 | 104953.7 | 30424.0 | 17664.2 | 1356.2 | 689.6 |
| Ratio (%) | 1.63 | 0.51 | 0.28 | 0.34 | 0.42 | 0.72 | 16.30 |

## C    PERPLEXITY ON TEST SETS

To more comprehensively demonstrate the effectiveness of our approach, we additionally use perplexity as an evaluation metric for comparative experiments. The experimental results in Table 6 indicate that our system (SimCAS) significantly enhances model performance compared to the baselines (BART).

## D    TIME LATENCY AT INFERENCE STAGE

At the inference stage, our framework additionally introduces a selection process compared with the standard transformer. Therefore, We study the proportion of our selector in the average inference time cost per sample. The results in Table 7 indicate that our selector only adds a small amount of time cost while significantly improving model performance, demonstrating the effectiveness of our approach.

## E    POTENTIAL TO SELECT REPRESENTATION

Because in the decoder cross-attention module, the encoded output from the encoder is used as "Key" and "Value" to participate in the calculation, its attention score can lead to the contribution of the corresponding token representation from encoded output to the current token decision at the decoding step. Figure5 demonstrates how the cross-attention scores change during the decoding of the reference output with one example in GovReport. We can observe that 1) most of the token decisions in the decoding phase focus only on a small set of encoded representations; 2) For each token decision, the contributions of different encoded token representations vary greatly. These phenomena suggest that there is still the feasibility of further filtering out low-contribution encoded representations.

## F    COMPATIBLE WITH SHORT-INPUT TASKS

While the sparse-attention transformers have been proven effective on a wide range of long-sequence datasets, as shown in Figure 6, these methods tend to underperform traditional full-attention transformers on the more common short-sequence tasks. However, in real scenarios, short-sequence inputs and long-sequence inputs are often mixed together, and the former occupies the vast majority. This limits the application scope of sparse-attention transformer architecture. In contrast, applying our framework SimCAS to existing full-attention transformers has strong flexibility. Specifically, given the full-attention model under SimCAS, if the input sequence exceeds the maximum length of a single chunk, we will perform chunking and selecting, otherwise, we can naturally switch to a standard short-text encoding form by skipping chunking procedures.

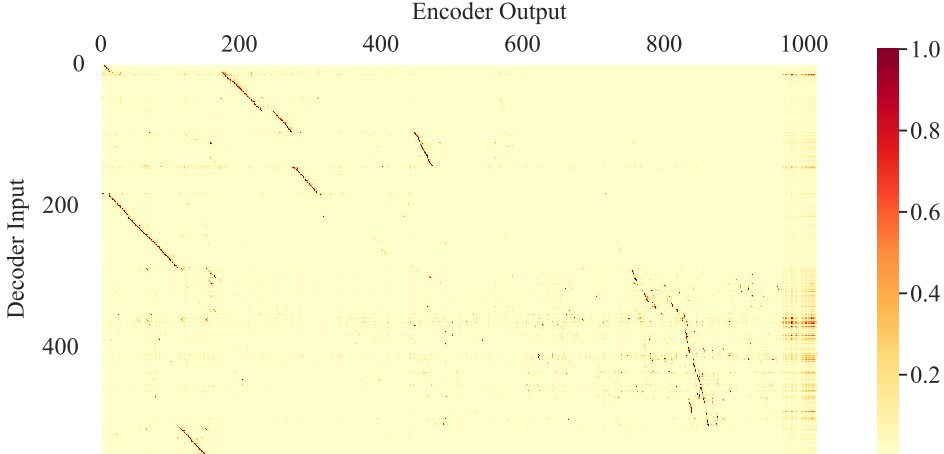

Figure 5: Visualization of the average cross-attention of all attention heads at all layers in the decoder (excluding start position). This example is generated by the BART$_{base}$+SimCAS trained on GovReport.

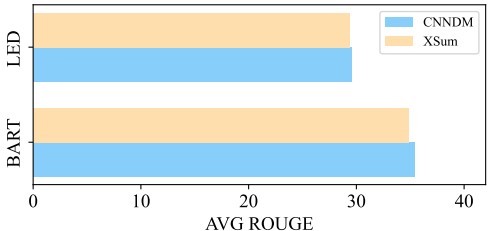

Figure 6: Comparison of the performance of full-attention PLM BART and sparse-attention PLM LED on short-text summarization datasets CNNDM (Nallapati et al., 2016) and XSum (Narayan et al., 2018). AVG ROUGE denotes the average of ROUGE-1/2/L F1 scores.

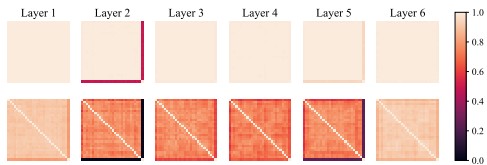

Figure 7: Evolution of similarities between start hidden states of each chunk after each encoding layer during one forward propagation. Only cosine similarity is used in the top half of the figure. The lower part takes into account both cosine similarity and Euclidean distance.

## G  EFFICACY OF SBA

As our framework introduces SBA to align inter-chunk information during the encoding steps, we explore the change of hidden state in the forward propagation process under the influence of this component. Figure 7 demonstrates the visualized similarity matrix between each chunk's start hidden state after the encoding block. It can be seen from an array of cosine similarity matrices in the top half that, through the alternation of SBA and encoding blocks, the cosine similarities between the start hidden states from different chunks are all close to 1, which indicates that their directions in high-dimensional space are almost the same.

Considering that cosine similarity can only reflect the closeness between vector directions, we design a similarity calculation method to add the measure of Euclidean distance. Given a pair of vectors $\boldsymbol{v}_1, \boldsymbol{v}_2 \in \mathbb{R}^d$, the custom similarity **Sim** between them is formulated as follows:

$$\mathbf{Sim} = \frac{(\boldsymbol{v}_1, \boldsymbol{v}_2)}{\|\boldsymbol{v}_1\| * \|\boldsymbol{v}_2\| * (1 + \|\boldsymbol{v}_1 - \boldsymbol{v}_2\|)},$$

by which, we scale the cosine similarity to observe its change. As can be seen from the lower part of Figure 7, after each encoder layer, the start hidden states will vary in scale due to different contexts.

## H  MORE EXPERIMENTAL DETAILS

**Input Structure** Since we aim to minimize dataset-specific modeling, we unify the processing of the used long-document summarization, multi-document summarization, and machine reading comprehension datasets. In the NarrativeQA dataset for machine reading comprehension, we concatenate together the question and reference content, each of which prepends a prefix to indicate the type of text.

**Training Details** In this paper, for the training of $\text{BART}_{\text{base}}$ on various datasets we uniformly employ the Adam optimizer with the following dynamic learning rate:

$$lr = \gamma \min(step^{-0.5}, step \times warmup^{-1.5}),$$

where $\gamma$ affects the maximum learning rate and *warmup* indicates the warmup steps, *step* is the number of updating steps, and $lr$ is the learning rate. In addition, there is another separate Adam optimizer for our reinforcement learning-based parameterized selector, in which the learning rate is fixed to $1 \times 10^{-4}$. The optimization of the selector and transformer is performed alternately, with one model trained while the other model is fixed.

**Inference Details** In practice, We also investigate how the model performs when the maximum effective input length is increased during inference. In Table 8, we can observe that although the maximum input length is set to 16384 during the training phase due to the memory limitation, increasing this maximum length during inference still improves the model performance.

**The Details of Selector** In our setting, the selector consists of an actor component and a critic component with PPO optimization, both of which are the feed-forward network, except for the final layers. In order to enable the selector to choose more diverse token representations instead of becoming homogeneous during the chunk-wise selection process, the state space consists of the current token representation and the selector's hidden state that is affected by previous actions. At the beginning of the selection procedure, the initial selector hidden state is the average of the start hidden state of all chunk representations. At each time step, we input the current selector hidden

Table 8: Performance of BART$_{base}$-SimCAS on WCEP with different maximum input lengths during inference. R-1/2/L is the ROUGE-1/2/L F1 score. BS refers to the model-based metric BERTScore. $\star$: the maximum input length in the training phase. $+\infty$: the input sequence to the model is complete. The best results are bolded.

| Maximum Input Length | R-1 | R-2 | R-L | BS |
|---|---|---|---|---|
| 4096 | 44.96 | 22.60 | 37.17 | 69.74 |
| 8192 | 45.41 | **22.79** | 37.40 | 70.51 |
| 16384$^\star$ | 45.22 | 22.48 | 37.34 | 70.46 |
| 32768 | 45.45 | 22.56 | 37.45 | 70.52 |
| $+\infty$ | **45.68** | 22.80 | **37.71** | **70.59** |

state and a chunk of token representations. Then the actor of the selector outputs a probability distribution over action space, and the critic of the selector outputs a single estimated scalar value for each token based on the hidden state selector and the corresponding token representation. For the state transition after executing current actions, we update the hidden state of the selector using the selected token representation from the current chunk. Note that in order to avoid the extreme case where the selector skips all tokens, in each chunk decision, if all actions are "*skipping*", they are all switched to "*selecting*", and the corresponding action probability and value are re-obtained. Additionally, to increase the training stability, the advantage in the PPO algorithm is approximated using the Generalized Advantage Estimation (GAE) (Schulman et al., 2016).

## I  CASE STUDY ON GOVREPORT DATASET

In addition to using regular automatic evaluation metrics to measure the effect of model generation, we also present some actual output to support the results. Figure 8 displays several examples of summaries generated by the fine-tuned base model BART$_{large}$ and our BART$_{base}$-SimCAS. We can observe that the system output of our model has fewer grammatical errors and higher coherence compared with the base model. Furthermore, since our model is able to perceive longer sequences, our output is more informative and better aligned with the reference text.

| System | Summary |
|---|---|
| Reference | in may 2018 , gao reported that the trust fund , which pays disability benefits to certain coal miners , faced financial challenges . the trust fund has borrowed from the u.s. treasury 's general fund almost every year since 1979 to make needed expenditures . gao 's june 2019 testimony included preliminary observations that coal operator bankruptcies were further straining trust fund finances because , in some cases , benefit responsibility was transferred to the trust fund . this testimony is based on gao 's report being released today , and describes ( 1 ) how coal mine operator bankruptcies have affected the trust fund , and ( 2 ) how dol managed coal mine operator insurance to limit financial risk to the trust fund . in producing this report , gao identified coal operators that filed for bankruptcy from 2014 through 2016 . gao analyzed information on commercially - insured and self - insured coal operators , and examined workers ' compensation insurance practices in four of the nation 's top five coal producing states . gao also interviewed dol officials , coal mine operators , and insurance company representatives , among others . coal mine operator bankruptcies have led to the transfer of about $ 865 million in estimated benefit responsibility to the federal government 's black lung disability trust fund ( trust fund ) , according to dol estimates . the trust fund pays benefits when no responsible operator is identified , or when the liable operator does not pay . gao previously testified in june 2019 that it had identified three bankrupt , self - insured operators for which benefit responsibility was transferred to the trust fund . since that time , dol 's estimate of the transferred benefit responsibility has grown — from a prior range of $ 313 million to $ 325 million to the more recent $ 865 million estimate provided to gao in january 2020 . according to dol , this escalation was due , in part , to recent increases in black lung benefit award rates and higher medical treatment costs , and to an underestimate of one company 's ( patriot coal ) future benefit claims . trust fund , filed from 2014 through 2016 dol 's limited oversight of coal mine operator insurance has exposed the trust fund to financial risk , though recent changes , if implemented effectively , can help address these risks . in overseeing self - insurance in the past , dol did not : estimate future benefit liability when setting the amount of collateral required to self - insure ; regularly review operators to assess whether the required amount of collateral should change ; or always take action to protect the trust fund by revoking an operators ' ability to self - insure as appropriate . in july 2019 , dol began implementing a new self - insurance process that could help address past deficiencies in estimating collateral and regularly reviewing self - insured operators . however , dol 's new process still lacks procedures for its planned annual renewal of self - insured operators and for resolving coal operator appeals should operators dispute dol collateral requirements . this could hinder dol from revoking operators ' ability to self - insure should they not comply with dol requirements . further , for those operators that do not self - insure , dol does not monitor them to ensure they maintain adequate and continuous commercial coverage as appropriate . as a result , the trust fund may in some instances assume responsibility for paying benefits that otherwise would have been paid by insurers . gao made three recommendations to dol to establish procedures for self - insurance renewals and coal operator appeals , and to develop a process to monitor whether commercially - insured operators maintain adequate and continuous coverage . dol agreed with these recommendations . |
| BART (1K) | of the eight coal mine operator bankruptcies gao identified from 2014 through 2016 , three resulted in a transfer of estimated benefit liability from the coal operator to the trust fund and five did not , according to dol . dol estimates for how these bankruptcies will affect the trust fund have considerably increased from what dol had previously reported . in june 2019 , gao reported that dol estimated that between $ 313 million to $ 325 million in benefit liabilities would transfer to the trust fund as a result of the bankruptcies . in january 2020 , however , dol provided updated estimates stating that $ 865 million in benefits would be transferred to the trust fund.gao also reported in june 2019 that the federal government does not have a reliable estimate of the amount of collateral dol required from coal mine operators to self-insure , and that benefit liabilities in excess of the collateral can be transferred from the bankrupt operator.what gao found the department of labor 's ( dol ) estimates for the extent of black lung benefit liabilities transferred from operators to the u.s. trust fund are considerably higher than dol 's estimates , and dol officials said that their estimates increased , among other reasons , to account for higher black lung award rates that occurred from fiscal years 2016 through 2019 , and higher medical treatment cost inflation in recent years.dol 's prior estimate for the patriot coal ( patriot ) bankruptcy did not fully cover the estimated benefit liabilities . when this occurs , benefit liabilities that in excess or in excess to the collateral could be transferred into the trust trust fund . for example , the collateral required from alpha natural resources ( alpha ) was about $ 12 million and approximately $ 494 million of estimated black lung liability transferred to a trust fund in 2016.since 2016 , several other self-insured operators have also filed for bankruptcy , including cambrian coal , cloud peak energy , murray energy , and westmoreland coal . dol does not estimate future benefit liability when setting collateral or regularly review operators to monitor their changing financial conditions . in the past , agency procedures require that collateral be obtained from operators in an amount deemed necessary and sufficient to secure the payment of the operators ' liability . to determine collateral amounts under the former process , the agency procedures stated that an operator 's net worth was to be equal to 3 , 5 , or 10 years of a operator 's annual black lung benefit payments made at the time of a coal operator 's self-insurance application depending on its net worth . specifically , if net worth were $ 1 billion or greater , then dol procedures set collateral equal to three years of benefit payments . if net worth is $ 1.1 billion , then net worth would equal to 5 , 10 , or 20 years of the operator 's monthly benefit payments , depending on the net worth of the underlying collateral . the amount of the required collateral from these operators would not have to be transferred in the future , and therefore would not be transferred.the federal government has not developed reliable estimates of future black lung benefits liability for the coal operators gao selected for review because dol has not established a time frame for doing so . in december 2017 , d dol announced that it was developing a new estimate of future benefit liabilities for coal operators , but it has not yet determined the extent to which the new estimate will address the limitations of the current estimate . without reliable estimates , it will be difficult for dol to determine whether future benefits liability will be transferred or whether additional actions may be needed to address these limitations . |
| SimCAS (16K) | coal mine operator bankruptcies resulted in a transfer of estimated benefit liability from the coal operator to the trust fund and five did not , according to the department of labor ( dol ) . dol estimates that between $ 313 million to $ 325 million in benefit liabilities would transfer to the trust fund as a result of these bankruptcies . gao was asked to review dol 's oversight of coal mine operators ' self - insurance . this report examines ( 1 ) the extent to which dol has taken steps to protect the financial interests of the trust fund and ( 2 ) the actions dol took to mitigate the financial losses of the bankrupt operators . to conduct this work , gao reviewed dol documents and interviewed dol officials . from 2014 through 2016 , three self - insured coal mine operations resulted in the transfer of $ 865 million of estimated black lung benefit liability to the u.s. trust fund ( trust fund ) . of the eight bankruptcies gao identified , three resulted in an transfer of the estimated benefit liabilities from the coal mine operators ( coal operators ) : three ( energy future holdings , peabody energy , and walter energy ) did not affect the trust fund . the amount of collateral dol required from these operators to self - insure did not fully cover their estimated benefits liabilities . when this occurs , benefit liabilities in excess of the collateral can be transferred to the trust fund . for example , the collateral dl required from energy future holdings was about $ 12 million and approximately $ 494 million of expected benefit liability transferred to the trust fund . dol also did not routinely consider potential future claims for which an operator could be responsible . in reviewing the most recent reauthorization memos for each of the self - insured operators , we found that these operators were not reauthorized since 1988 . in january 2020 , dol provided updated estimates stating that $ 865 million in benefits would be transferred from these two bankruptcies as a consequence of the three other bankruptcies , but dol does not have procedures that specify the duration of an operator 's self insurance authority or the conditions under which that authority would not be renewed . in the absence of a process to monitor operator compliance with program requirements , the agency risks not identifying or cancelling operator coverage . in addition , the new procedures do not specify , among other things , how long an operator will be authorized to self insure ; when an operator is authorized to do so ; when the operator is not renewing the operator 's authority ; and what conditions are under which this authority would be not renewed . the new requirements for setting collateral and for the more frequent review of self insured operators are key elements of internal controls , which call for agency management to implement control activities through policy . however , dl 's new self - insurance procedures do not specify , for example , how much collateral the agency will require an operator to secure and how much time dol appeals decisions should be made . in particular , dls does not specify a goal for the amount of time that it will take for the operator to apply for and receive a renewal application . without such a goal , it is difficult for dol to ensure that the application is valid and that the operator has a valid application for and receives an extension of the application . in commenting on a draft of this report , dod concurred with gao 's findings and recommendations . |

Figure 8: Example summary generated by BART and SimCAS trained on GovReport dataset. The maximum input length of standard BART and SimCAS is 1024 and 16384 respectively. The sentence in green is included in the SimCAS summary, while the one in red is discarded.

