# OpenReview forum: "Chunk, Align, Select: A Simple Long-sequence Processing Method for Transformers"
_ICLR.cc/2024/Conference — ICLR 2024 Conference Withdrawn Submission_

### Official Review · Reviewer_pFUP · 2023-10-31

**Soundness:** 3 good
**Presentation:** 3 good
**Contribution:** 3 good
**Rating:** 6
**Confidence:** 3

**Summary:**

The paper develops a way to use pre-trained transformers efficiently on long sequences without the need for further pre-training due to architectural changes.

The method first creates non-overlapping chunks to apply Transformer attention locally only (within chunks). After each layer, it does an elementary form of inter-chunk interaction by updating each bos and eos special tokens (indicating chunk boundaries within each chunk) with the average of newly produced eos and bos tokens (after the earlier local chunk processing layer) respectively.

The method uses reinforcement learning for token selection for decoding with rewards created to penalize selecting too many tokens and other rewards based on language modeling probabilities. PPO is used in an actor-critic framework.

**Strengths:**

1. While the literature is saturated with efficient transformers for handling long sequences, this work can be applied to pre-trained models like BART directly without additional pre-training.

2. The performance is quite decent compared to baselines.

**Weaknesses:**

The method could be counted as a form of dynamic pruning technology. There are already several works in that area. For example, transkimmer [1] already has dynamic token pruning. And there are other works in similar directions [2]. The usability of RL for token pruning is not a particularly surprising method, and chunking + local attention is a very standard policy for efficiency gain. More research can be done in contextualizing the work in the literature review by exploring other related works to [1,2] in the citation network. That said, The inter-chunk communication through average is interesting (although quite simple and could be seen as a hack)  and the overall synthesis seems to work well.


[1] Transkimmer: Transformer Learns to Layer-wise Skim - Guan et al. ACL 2022

[2] Learned Token Pruning for Transformers - Kim et al. KDD 2022

**Questions:**

1. Is the chunking method only applied to the encoder? Would there be any problem in applying it to the decoder and decoder-only models (standard LLMs) after appropriate changes (such as causality constraint in averaging for chunk alignment)?

2. Is there a graph for per-iteration speed-up comparisons with baseline?

---

> ### Author Response · Authors · 2023-11-21
> **Official Comment by Authors**
>
> Thank you for the valuable reviews, here are our replies.
>
> > The method could be counted as a form of dynamic pruning technology. There are already several works in that area. For example, transkimmer [1] already has dynamic token pruning. And there are other works in similar directions [2]. The usability of RL for token pruning is not a particularly surprising method, and chunking + local attention is a very standard policy for efficiency gain. More research can be done in contextualizing the work in the literature review by exploring other related works to [1,2] in the citation network. That said, The inter-chunk communication through average is interesting (although quite simple and could be seen as a hack) and the overall synthesis seems to work well.
> >
> > [1] Transkimmer: Transformer Learns to Layer-wise Skim - Guan et al. ACL 2022
> >
> > [2] Learned Token Pruning for Transformers - Kim et al. KDD 2022
>
> As mentioned in the related works, token pruning is a common technique used to reduce the overhead of training and inference. However, most existing methods are only suitable for short sequence tasks and are often not applicable to the case of long sequence inputs. In contrast to previous work, we propose a selection strategy in the context of chunking to choose hidden representations, significantly improving model performance. Although the use of RL for token pruning has been explored, these methods typically only use the final output signal (such as cross-entropy loss in TR-BERT [1]) as a coarse-grained reward for training the selection strategy. In our method, we additionally utilize attention signals to design the more fine-grained reward for each action. Our experiment results demonstrate the effectiveness of our approach.
>
> [1] Ye D, Lin Y, Huang Y, et al. Tr-bert: Dynamic token reduction for accelerating bert inference[J]. arXiv preprint arXiv:2105.11618, 2021.
>
> > Is the chunking method only applied to the encoder? Would there be any problem in applying it to the decoder and decoder-only models (standard LLMs) after appropriate changes (such as causality constraint in averaging for chunk alignment)?
>
> In our paper, the chunking method is only used for the encoder. The method we proposed in the paper does not adapt directly to the decoder-only model. The key issues are how to conduct the inter-chunk communication in the decoder-only model and how to design rewards using decoder self-attention.
>
> > Is there a graph for per-iteration speed-up comparisons with baseline?
>
> According to your suggestion, we provide these graphs for per-iteration speed-up comparison with SLED on all our test sets in the supplementary materials. From these graphs, we can see that compared to the previous chunking method SLED, our method significantly reduces the time overhead. At the same time, the experimental results in the paper also show that the model performance has greatly improved.

---

### Official Review · Reviewer_9Rva · 2023-10-31

**Soundness:** 4 excellent
**Presentation:** 3 good
**Contribution:** 4 excellent
**Rating:** 8
**Confidence:** 4

**Summary:**

This paper introduced a Simple learning framework with three typical operations: Chunk, Align, and Select to enable pre-trained encoder-decoder language models to process long-context. The chunking process is a variant of batchfying. The aligning process is the forwarding via BART or other LM encoders. The proposed selector network as well as the RL-based training is novel and interesting. The experiments are very solid, which covers almost all long-context datasets as far as I know.

**Strengths:**

1. The experiments are comprehensive and the results are robust and strong. The evaluation tasks and datasets coverage is excellent.  I appreciate that the authors conduct experiments on NarrativeQA to demonstrate the method's effectiveness and the NarrativeQA is recognized as the hardest benchmark for long-context transformer.

2. The method is intuitive and easy to understand. Even if the chunking and selection/retrieval is not that novel, the introduced PPO-based training for selector network is interesting and brings some insights.

**Weaknesses:**

1. The computational timecost brought by retrieval or selection based methods are always a significant issue. I did not find such discussions in main context and appendix about the latency part. The author should discuss and measure it to show the tradeoff between performance gain and latency increase.

2. As the selector FFN is randomly initialized but the backbone is pre-trained well, I think in the first 10k training iterations, the training might be unstable if following the alternative updates on a well-trained model and a newly-initialized model. The author provides limited details about this.

3. The ablation study on the chunk length might be interesting and important. A smaller chunk length brings better granularity and a larger chunk length accelerates the inference. Is the selected 512 length the best length for chunks?

**Questions:**

1. For footnote 1 of Page 3, how you deal with cases that a sentence is longer than a chunk as sometimes the sentence segmentation tool does not work well to split the sentence to desired length?

2. The selector is a small-size 8M FFN. Did you consider to scale up the parameters and also change the FFN to a RoBERTa model with binary classification head? This might be better for the action decision.

3. If I understands the paper well, the selector is not pre-trained and is only randomly initialized FFN. Then you fine-tune the BART and selector together on specific task. Did you try to freeze BART and pre-trained the FFN selector on the same pre-training corpus of BART? Personally, I think if your selector FFN is only 8M parameters, the pre-training might not be helpful. But if the selector size is scaled up, this might be good to the model.

4. Some missing related works: TRIME (Zhong et. al., 2023) and LongMem (Wang et. al., 2023) are related to this method in terms of chunking and memorizing; Landmark Attention (Mohtashami and Jaggi, 2023) is related to this method in terms of S and E tokens.

**Details Of Ethics Concerns:**

None.

---

> ### Author Response · Authors · 2023-11-21
> **Official Comment by Authors (Part 1)**
>
> Thank you for the valuable reviews, here are our replies.
>
> > The computational timecost brought by retrieval or selection based methods are always a significant issue. I did not find such discussions in main context and appendix about the latency part. The author should discuss and measure it to show the tradeoff between performance gain and latency increase.
>
> At the inference stage, the average time overhead of the selection process (Select) and the whole process (All), and their ratio (Ratio)  for a single sample are shown below. In addition, it should be noted that in our experiments, we used the V100 which has a relatively slow computation speed. If more advanced GPUs are used, the latency will be significantly reduced.
>
> |                | arXiv  | PubMed  | GovReport | SummScreen | Multi-News |  WCEP  | NarrativeQA |
> | :------------: | :----: | :-----: | :-------: | :--------: | :--------: | :----: | :---------: |
> | Selection (ms) | 157.5  |  70.4   |   293.9   |   103.4    |    74.2    |  9.8   |    112.4    |
> |    All (ms)    | 9661.5 | 13799.1 | 104953.7  |  30424.0   |  17664.2   | 1356.2 |    689.6    |
> |   Ratio (%)    |  1.63  |  0.51   |   0.28    |    0.34    |    0.42    |  0.72  |    16.30    |
>
> > As the selector FFN is randomly initialized but the backbone is pre-trained well, I think in the first 10k training iterations, the training might be unstable if following the alternative updates on a well-trained model and a newly-initialized model. The author provides limited details about this.
>
> Actually, we were surprised to find that the transformer’s MLE loss with label smoothing decreases quite normally during the training phase in most cases. We believe this may be because, although the selector is randomly initialized, it only participates in the selection of the transformer’s hidden representations and does not severely disrupt the representation information. Therefore, the training of the transformer is relatively stable. We have submitted the figures of MLE loss changes during the training process to the supplementary materials and included it in the appendix of the paper.
>
> > The ablation study on the chunk length might be interesting and important. A smaller chunk length brings better granularity and a larger chunk length accelerates the inference. Is the selected 512 length the best length for chunks?
>
> In our experiments, we found that with the same input length, chunk sizes 512 and 1024 have similar model performance, and the model performance significantly decreases when the chunk size is 256. Therefore, in order to process longer sequences while maintaining sound model performance, we set the chunk size to 512.
>
> > For footnote 1 of Page 3, how you deal with cases that a sentence is longer than a chunk as sometimes the sentence segmentation tool does not work well to split the sentence to desired length?
>
> In fact, although it is rare, we have indeed observed some sentences that are longer than the chunk size. In this case, we will directly discard the part of the sentence that exceeds the chunk size.

---

> ### Author Response · Authors · 2023-11-21
> **Official Comment by Authors (Part 2)**
>
> > The selector is a small-size 8M FFN. Did you consider to scale up the parameters and also change the FFN to a RoBERTa model with binary classification head? This might be better for the action decision.
>
> In our method, it is feasible to improve the model performance by increasing the scale of the parameterized selector. However, considering the time overhead brought by the selection process, we hope that the increased overhead is small and the system still has good performance. In my opinion, changing FFN to RoBERTa can improve the model performance, but the time overhead of the selection process may be large.
>
> > If I understands the paper well, the selector is not pre-trained and is only randomly initialized FFN. Then you fine-tune the BART and selector together on specific task. Did you try to freeze BART and pre-trained the FFN selector on the same pre-training corpus of BART? Personally, I think if your selector FFN is only 8M parameters, the pre-training might not be helpful. But if the selector size is scaled up, this might be good to the model.
>
> Because we consider the compatibility of our method with existing pretrained language models as one of the contributions, we did not conduct additional pre-training. Theoretically, the current selector’s model size is relatively small, making it difficult to achieve good generalization through pre-training. However, pre-training a larger scale selector may improve the model effect, but the time latency will also increase.
>
> > Some missing related works: TRIME (Zhong et. al., 2023) and LongMem (Wang et. al., 2023) are related to this method in terms of chunking and memorizing; Landmark Attention (Mohtashami and Jaggi, 2023) is related to this method in terms of S and E tokens.
>
> Thank you for bringing these papers to our attention. We appreciate your suggestion and agree that these works are indeed relevant to our study. We will review these papers and discuss their relation to our method in the revised manuscript.

---

### Official Review · Reviewer_5A1y · 2023-11-01

**Soundness:** 3 good
**Presentation:** 3 good
**Contribution:** 2 fair
**Rating:** 3
**Confidence:** 4

**Summary:**

This paper presents an approach concentrated on the refinement of long-sequence modeling. The proposed method divides a sequence into a series of chunks. This technique meticulously aligns the inter-chunk information during the encoding phases, and subsequently, the most pivotal hidden states are discerningly selected from the encoder to facilitate the decoding process. Experiment results show that in long-text abstractive summarization and reading comprehension tasks, the proposed method outperforms strong baselines of long-sequence processing.

**Strengths:**

1. The authors propose a simple framework that can directly be used on existing PLMs for processing long sequences.
2. The authors propose a RL method to facilitate the transformer to concentrate more effectively on the crucial encoded hidden states.
3. Experiments show better results over strong baselines.

**Weaknesses:**

1. Missing related work. Long sequence Transformer is a hot topic, including two main directions: efficient computation or length extrapolation (train-short-test-long). In the realms of long-sequence Transformers, there appears to be a noticeable omission in the exploration of length extrapolation. Here are two related studies:
   1) A Length-Extrapolatable Transformer
   2) Train Short, Test Long: Attention with Linear Biases Enables Input Length Extrapolation
2. The proposed RL objectives lack solid motivation. It is well-known that RL objectives are hard to implement due to the variance of reward scores. There lacks necessary evidence to show how these RL objectives work and whether these RL objectives are necessary (compared to traditional MLE loss).
3. The authors only conduct experiments on BART models.  It remains unclear whether the proposed method still works on more recent models, like LLAMA.

**Questions:**

1. How about the PPL scores on long text modeling? It is a widely-used metric to evaluate the performance of  long-text language modeling.
2. A minor stylistic observation pertains to the line spacing within the background section.

---

> ### Author Response · Authors · 2023-11-21
> **Official Comment by Authors**
>
> Thank you for the valuable reviews, here are our replies.
>
> > Missing related work. Long sequence Transformer is a hot topic, including two main directions: efficient computation or length extrapolation (train-short-test-long). In the realms of long-sequence Transformers, there appears to be a noticeable omission in the exploration of length extrapolation. Here are two related studies:
> >
> > 1. A Length-Extrapolatable Transformer
> > 2. Train Short, Test Long: Attention with Linear Biases Enables Input Length Extrapolation
>
> We thank the reviewer for these provided related studies and will modify the manuscript to include the content for the length extrapolation.
>
> > The proposed RL objectives lack solid motivation. It is well-known that RL objectives are hard to implement due to the variance of reward scores. There lacks necessary evidence to show how these RL objectives work and whether these RL objectives are necessary (compared to traditional MLE loss).
>
> In our method, we use a transformer as a simulation environment to optimize a reinforcement learning selection strategy for selecting crucial token hidden representations. We design rewards using the attention and output signals of the transformer, and obtain and update states using the transformer’s hidden representation. Due to the lack of ground truth to determine the importance of hidden representations, and the tendency of attention scores to shift as training progresses, we believe that reinforcement learning methods, especially these algorithms like PPO that can maintain the exploration characteristics, can handle this situation well. In addition, since the “Chunk” stage transforms long sequences into multiple blocks, selecting from one block at a time step can also be intuitively seen as a Markov Decision Process. In our paper, there are many empirical results that prove the effectiveness of our RL selection strategy. For example, in Table 4, we observed that adding the “Select” module can greatly improve the performance of the model. In fact, in our early research, we did not have an RL selection module, and the performance of the model at that time was worse than the current strong baselines. In addition, in Figure 3, from the left graph, we can see that with the progress of training, the number of selected hidden representations first drops rapidly, and then converges to around our preset value (2048). From the right graph, we can see that even though the input length at the inference stage is exponentially increased by adding pad tokens, the number of selected hidden representations still grows slowly. This indicates that our selector has the ability to filter out low-contribution representations. Based on these results, we believe that our RL objective is effective and necessary.
>
> > The authors only conduct experiments on BART models. It remains unclear whether the proposed method still works on more recent models, like LLAMA.
>
> Our method is applicable to most pre-trained encoder-decoder models, including BART, PEGASUS, T5, LED and PRIMERA. For decoder-only models like LLAMA, we believe that although our method cannot be applied directly, it still has high reference value.
>
> > How about the PPL scores on long text modeling? It is a widely-used metric to evaluate the performance of long-text language modeling.
>
> Following recent long-sequence processing work such as Unlimiformer [1], we use commonly used metrics such as ROUGE and BERTScore in specific downstream tasks for more persuasive model performance comparisons. Thank you for your suggestion, we have added the PPL results on the test set.
>
> |                        | arXiv | PubMed | GovReport | SummScreen | Multi-News | WCEP | NarrativeQA |
> | ---------------------- | ----- | ------ | --------- | ---------- | ---------- | ---- | ----------- |
> | BART$_{base}$          | 3.46  | 3.55   | 3.74      | 3.89       | 3.90       | 3.93 | 3.15        |
> | BART$_{base}$ + SimCAS | 1.93  | 1.72   | 1.83      | 2.38       | 2.11       | 2.18 | 1.55        |
>
> [1] Bertsch A, Alon U, Neubig G, et al. Unlimiformer: Long-range transformers with unlimited length input[J]. arXiv preprint arXiv:2305.01625, 2023.
>
> > A minor stylistic observation pertains to the line spacing within the background section.
>
> We will review the line spacing in the background section and make necessary adjustments to improve readability and conform to the formatting guidelines. Your attention to detail is much appreciated.

---

> > ### Author Response · Authors · 2023-11-23
> > **Correctness for Perplexity**
> >
> > We apologize for the omission in our perplexity calculation. The current result represents the exponent in 'exp(i)'. The actual perplexity is as follows:
> > |                        | arXiv | PubMed | GovReport | SummScreen | Multi-News | WCEP | NarrativeQA |
> > | ---------------------- | ----- | ------ | --------- | ---------- | ---------- | ---- | ----------- |
> > | BART$_{base}$          | 31.82  | 34.81   | 42.10      | 48.91      | 49.40    | 50.91 | 23.34        |
> > | BART$_{base}$ + SimCAS | 6.89  | 5.58  | 6.23     | 10.80     | 8.25      | 8.85 | 4.71       |

---

### Official Review · Reviewer_mi4g · 2023-11-04

**Soundness:** 3 good
**Presentation:** 3 good
**Contribution:** 2 fair
**Rating:** 5
**Confidence:** 4

**Summary:**

This paper induces attention sparsity by selecting the positions of Key and Value in \textbf{cross-attention}.  The proposed method first chunks a sequence into blocks, then aligns the bos and eos of each block by using the average of them in every block of the next layer, in the last, it filters the positions of each block according to language modeling likelihood. This paper experiments on many summarization datasets and improves the baseline model Bart by a large margin. The main improvement comes from chunking and selecting.

**Strengths:**

1. This paper shows that sparsification in cross-attention has a surprising potential for performance improvement, the proposed method improves about 20% over baselines.

**Weaknesses:**

1. The proposed method depends on cross-attention, and we could not introduce it to the encoder-only or decoder-only Transformer model.   If it can outperform existing LLMs, this would not be a weakness.

2. Experiments on other tasks are limited. This paper mainly experiments on summarization tasks, but does not experiment on CNN/DM or Xsum, which are the most compared data. For other tasks, this paper only does Narrative QA.  More experiments on document translation or classification would be an advantage.

**Questions:**

1. Do the baselines sparse self-attention?

2. There is a typo in Figure 3, the blue line should be selected length.

3. Table 4 should add a row of pure baseline, i.e., w/o neither. It is a surprise that we need both chunk and select, or we will lose the baseline.

4. I would like to see a comparison regarding latency.

---

> ### Author Response · Authors · 2023-11-21
> **Official Comment by Authors**
>
> Thank you for the valuable reviews, here are our replies.
>
> > The proposed method depends on cross-attention, and we could not introduce it to the encoder-only or decoder-only Transformer model. If it can outperform existing LLMs, this would not be a weakness.
>
> In our experiment, we utilized the encoder-decoder transformer BART as the backbone. This choice was made because, among pre-trained language models of a relatively smaller scale, this type of transformer architecture often exhibits superior performance in downstream tasks. Simultaneously, our method is well-suited to the characteristics of the encoder-decoder transformer, resulting in a low degree of coupling between our method and this type of transformer. For encoder-only and decoder-only transformers, our method may not be applied directly, but it still holds sound reference value for  long-sequence processing of encoder-only model and decoder-only model, and the environment construction of RL-based selection policy.
>
> > Experiments on other tasks are limited. This paper mainly experiments on summarization tasks, but does not experiment on CNN/DM or Xsum, which are the most compared data. For other tasks, this paper only does Narrative QA. More experiments on document translation or classification would be an advantage.
>
> The CNN/DM and XSum datasets are commonly used for text summarization tasks. However, the average input text length for the former is 766.1, while for the latter it is 430.2. Therefore, standard full-attention models can already handle them well on consumer-grade GPUs, eliminating the need to introduce long sequence processing techniques. Based on your suggestions, we have incorporated additional experiments on the IMDB document classification dataset. The results are presented as follows.
>
> |   Method   | Accuracy |
> | :--------: | :------: |
> |  RoBERTa   |   95.0   |
> |  BIGBIRD   |   95.2   |
> | Longformer |   95.7   |
> |   SimCAS   |   96.1   |
>
> > Do the baselines sparse self-attention
>
> In the baselines we demonstrated, LED, PRIMERA, and BIGBIRD have implemented sparse self-attention in encoder module.
>
> > There is a typo in Figure 3, the blue line should be selected length.
>
> Thank you for pointing out this issue. We will correct this error in the new version.
>
> > Table 4 should add a row of pure baseline, i.e., w/o neither. It is a surprise that we need both chunk and select, or we will lose the baseline.
>
> In Table 4, we demonstrate the contribution of each part in SimCAS to the performance improvement when the backbone is BART. If both the “Chunk” and “Select” modules are removed, the model will degrade to the standard BART. The relevant experimental results have already been shown in BART+Standard (Table 1 2 3):
>
> > I would like to see a comparison regarding latency.
>
> Based on a V100 and the backone BART$_{base}$, we tested the average inference time (ms) of a single sample in each test set for SLED and SimCAS, and the results are as follows:
>
> | Method |  arXiv  | PubMed  | GovReport | SummScreen | Multi-News |  WCEP  | NarrativeQA |
> | :----: | :-----: | :-----: | :-------: | :--------: | :--------: | :----: | :---------: |
> |  SLED  | 28375.3 | 25323.0 | 153560.5  |  59012.7   |  37253.0   | 3120.6 |   1711.5    |
> | SimCAS | 9661.5  | 13799.1 | 104953.7  |  30424.0   |  17664.2   | 1356.2 |    689.6    |

---

### Author Response · Authors · 2023-11-23
**Official Comment by Authors**

Dear Reviewers,

As the time for author-reviewer discussion is soon to close, we are looking forward to your valuable feedback on our responses. We are also more than willing to address any further concerns you might have.

Awaiting your response with anticipation and gratitude.

Best wishes,

The Authors

---

### Author Response · Authors · 2023-11-23
**Official Comment by Authors**

Dear Reviewers, ACs and SACs,

We are very grateful for the detailed and valuable suggestions from all the reviewers to help improve the quality of our paper. We will do our best to address the issues raised by the reviewers. Currently, based on the reviews, we have added relevant citations and experiment results to the manuscript, and the latest version has been submitted.

We do sincerely appreciate you, and cheers!

Authors

---

### Meta-Review · Area_Chair_gqL3 · 2023-12-06

**Metareview:**

The paper proposes a method for enabling Transformers to deal with long sequence lengths through chunking. RL is used to learn to select representative tokens per chunk in the encoder, which are used in cross-attention with the decoder. Results show improvements over previous long context methods for comparable base models. The main weaknesses are that the approach as applied is limited to encoder-decoder models, and the experiments are limited to a few summarization datasets. Not all the results are directly comparable due to different base models. The paper will therefore be strengthed with further experiments and improvements in the clarity of the presentation.

**Justification For Why Not Higher Score:**

While the paper has some positive results, the contribution and experimental evaluation is limited.

**Justification For Why Not Lower Score:**

N/A

---

### Decision · Program_Chairs · 2024-01-16

Reject